# Beam Steering Technology of Optical Phased Array Based on Silicon Photonic Integrated Chip

**DOI:** 10.3390/mi15030322

**Published:** 2024-02-26

**Authors:** Jinyu Wang, Ruogu Song, Xinyu Li, Wencheng Yue, Yan Cai, Shuxiao Wang, Mingbin Yu

**Affiliations:** 1National Key Laboratory of Materials for Integrated Circuits, Shanghai Institute of Microsystem and Information Technology, Chinese Academy of Sciences, 865 Changning Road, Shanghai 200050, China; 2University of Chinese Academy of Sciences, Beijing 100049, China; 3Shanghai Industrial μTechnology Research Institute, Shanghai 201800, China

**Keywords:** beam steering, optical phased array, silicon photonics, divergence angle, scanning range

## Abstract

Light detection and ranging (LiDAR) is widely used in scenarios such as autonomous driving, imaging, remote sensing surveying, and space communication due to its advantages of high ranging accuracy and large scanning angle. Optical phased array (OPA) has been studied as an important solution for achieving all-solid-state scanning. In this work, the recent research progress in improving the beam steering performance of the OPA based on silicon photonic integrated chips was reviewed. An optimization scheme for aperiodic OPA is proposed.

## 1. Introduction

LiDAR is a sensing technology that obtains information about the position, velocity, and other characteristics of a target object by emitting laser and processing the returned information [1]. Compared to traditional radars such as microwave radar and millimeter wave radar, LiDAR can greatly improve the ranging accuracy and expand the scanning range of the radar. It also has advantages, such as being less affected by environmental lighting [2] and having good resistance to electromagnetic interference. Therefore, LiDAR is widely used in autonomous driving [3], imaging [4,5,6,7], remote sensing surveying [8], and space communication [9,10,11]. To meet the needs of free-space optical communication in fields such as autonomous driving and drones, LiDAR should have a wide range of scanning angles and long-range detection capabilities [12,13]. In order to distinguish objects, the beam emitted by LiDAR should also have high resolution. When applied in fields such as remote sensing measurement and imaging, the higher resolution can achieve more detailed measurements and present more elaborate images [14]. Nowadays, research on LiDAR focuses on expanding the scanning field of view (FOV), improving the scanning resolution, and extending the detection distance, generally moving towards miniaturization and integration.

LiDAR can be divided into three categories based on whether they contain mechanical components: mechanical LiDAR [15], hybrid solid-state LiDAR [16], and all-solid-state LiDAR [17]. OPA is a device that achieves all-solid-scanning by controlling the laser wavefront through the phase. Compared with mechanical LiDAR and hybrid solid-state LiDAR, there are no mechanical components in the OPA. OPA can achieve stable, accurate, fast beam deflection and arbitrary beam pointing [11,18], which can improve the robustness of LiDAR equipment. The current research on 1-D OPA is already mature and there are also excellent results on 2-D OPA [19,20,21,22,23,24]. 1-D OPA is divided into two mechanisms: end emission and grating emission. Compared with OPA based on liquid crystals (LC) and a micro-electromechanical system (MEMS), OPA based on silicon photon platforms can achieve a larger scanning range and faster scanning speeds [18,21,25,26,27]. In addition, OPA based on the photonic integrated chip (PIC) is compatible with complementary metal-oxide semiconductor (CMOS) technology and can be fabricated on a large scale by highly mature CMOS technology with the advantages of high production and low cost [28,29]. With the development of silicon-based optoelectronics in recent years, such as the development of CMOS compatible germanium–silicon or silicon defect photodetectors, there is an opportunity to integrate OPA with low-cost receivers on chips [30,31,32,33,34,35]. This article reviews the research progress of OPA based on PIC in recent years, mainly focusing on the grating emission 1-D OPA. In the Section 2, we introduce the parameters affecting beam quality. In Section 3, we elaborate on the schemes and progress for expanding the scanning range and improving resolution on the longitudinal and transversal scanning dimensions. To expand the scanning range, we propose an optimization scheme for aperiodic OPA. The unidirectionality that affects the output efficiency of OPA is also introduced in detail.

## 2. Scanning Principle of PIC OPA

OPA consists of splitter tree, phase shifters, and emitting array as shown in Figure 1. The incident light is divided into several channels through a cascaded splitter tree, and the phase of the light in each channel is modulated by a phase shifter. Finally, the light enters the emission array and radiates outward, forming a scanning beam in the specified direction. This direction is determined by the structure of the emitting elements, the wavelength of the input light, and the relative phase among different channels [36,37].

The emitting array is a crucial device for beam scanning, consisting of several waveguide grating antennas (WGAs). The typical structure of WGA is shown in Figure 2.

The effective refractive index of the WGA is modulated periodically through the periodic grating [38]. When passing through the grating region, the guided mode in the WGA excites a diffraction field and emits energy to free space through diffracted light. The optical path difference between two beams diffracted by adjacent periodic gratings satisfies the following formula [39]:(1)$$n_{eff}-n_{c}sin\theta =\frac{\lambda }{\Lambda },$$

The maximum value of diffracted light appears in the *θ* direction. Where *neff* and *nc* are the effective refractive index of the waveguide fundamental mode and the refractive index of cladding, respectively. *Λ* is the period of the WGA and *λ* is the wavelength of the incident light. *θ* is defined as the angle between the emitted light and the vertical direction, which is the longitudinal scanning angle of OPA; the range of *θ* modulated by the wavelength is called the longitudinal scanning range of OPA. The near-field electric field intensity of periodic structured gratings follows an exponential decay of *e***^−^**^2*αx*^, where *α* is the perturbation intensity of the grating and *x* is the direction of light propagation in the WGA. The effective emission length of a WGA is the length at which the diffraction intensity decays to *e*^−2^ of its initial value, that is *L* = 1/*α* [40]. When the radiation intensity remains consistent, the longitudinal divergence angle of the beam is expressed by the following formula [39]:(2)$$\Delta \theta \approx \frac{0.886\lambda }{N\Lambda cos\theta }\left(rad\right),$$

The ratio of the upward part of the light emitted from the WGA to the sum of the up and down directions is called unidirectionality and is described as follows [41]:(3)$$D=\frac{P_{up}}{P_{up}+P_{down}},$$

Ignoring the longitudinal scanning of OPA and only considering the transverse scanning determined by the phase relationship of array emitters, the scanning principle of OPA can be described by a 1-D periodic OPA with N units as shown in Figure 3.

Assuming that the directional function of the emitters is isotropic (i.e., the directional factor *f* = 1), the amplitude of the emitters is *A*, the interval between emitters is *d*, and the phase of each emitter unit is (*i* − 1) Δ*φ*. At an observation point *r*_0_ away from OPA, the field intensity is as follows:(4)$$E=\sum_{i=1}^{N}Afe^{-j(i-1)\Delta \varphi }\frac{e^{-j\frac{2\pi }{\lambda }r_{i}}}{r_{0}},$$

When the observation point is far from the optical phased array, the distance from the observation point to each array element can be approximated by *r*_0_. However, when considering the phase, approximation cannot be used. The distance from the observation point to each array element is *r_i_*. When *d* < $\frac{\lambda }{2}$, the OPA can achieve aliasing-free beam steering with 180° FOV in the transversal direction of the far field. When *d* > $\frac{\lambda }{2}$, periodic grating lobes appear in the far field. When the main beam deflects to the angle where a grating lobe with an amplitude equal to it appears, the range is defined as the FOV range of the periodic OPA and $\Phi_{FOV}=2arcsin(\frac{\lambda }{2d})$. The transversal divergence angle of the beam is expressed by the following formula [42]:(5)$$\Delta \Phi \approx \frac{0.886\lambda }{Ndcos\theta }\left(rad\right),$$

According to Equation (5), the divergence angle of a beam is inversely proportional to the aperture of the OPA (*Nd*). Therefore, for an OPA with uniform spacing, when the spacing between emitters is fixed, the divergence angle of the beam can be reduced by increasing the number of emitters.

## 3. Schemes and Review for Improving OPA Device Performance

This section summarizes the technical solutions for improving the beam quality of OPA in both longitudinal and transverse dimensions. In Section 3.1, we review the solutions to improve the quality of longitudinal scanning by expanding the scanning range and reducing the divergence angle. In Section 3.2, we classify the OPA into periodic distribution OPA and aperiodic distribution OPA. The transversal scanning range and divergence angle in these two technical schemes are also reviewed.

### 3.1. Improving Beam Quality of Longitudinal Dimension

#### 3.1.1. Expanding Scanning Range of Longitudinal Dimension

According to Section 2, the longitudinal scanning angle of OPA is determined by the structure of the WGA and the wavelength of incident light, with a resolution of Δ*θ* determined by the effective emission length of the WGA. For WGA with a determined structure, the longitudinal scanning angle of OPA varies approximately linearly with the working wavelength [43,44], and the longitudinal scanning range is also limited due to the limited working bandwidth of the light source. An OPA expanding the scanning range by polarization multiplexing was proposed in 2021 [45]. The OPA switches the input light between TE_0_ mode and TM_0_ mode through cascading an MZI with a polarization splitter rotator (PSR). The grating is formed by a 340 nm thick Si waveguide with an etching depth of 70 nm. By reasonably designing the cross-sectional size of the waveguide, the difference in effective refractive indices between TE_0_ mode and TM_0_ mode in the waveguide is reduced, and ultimately achieves a longitudinal continuous scanning of 28.2° within the bandwidth of 1500–1600 nm. In 2022, Zhao et al. added an optical switch in front of the OPA to control the forward/reverse input of light in the array [46]. Combined with the polarization multiplexing scheme described in [45], a continuous scanning range of 54.5° was achieved on the 340 nm SOI platform. In the same year, a polarization multiplexed OPA based on a 220 nm SOI platform was demonstrated as shown in Figure 4a [47]. Due to the significant difference in effective refractive indices between TE_0_ mode and TM_0_ mode in a 220 nm thick single-mode Si waveguide, although a longitudinal scanning range of two times was achieved during the experiment, continuous scanning could not be achieved. Zhao et al. proposed in 2023 a polarization multiplexing OPA in which TE_0_ and TM_0_ modes are transmitted in two staggered arrays [48]. The light is switched to the TE_0_ or TM_0_ mode by the optical switch after coupling to the OPA. A superlattice waveguide grating composed of two WGAs with different widths then forms an array. After the TE_0_ or TM_0_ mode, light passes through power splitters; it passes through the PSR and enters its respective array, achieving a 28° scanning range. The research and design of PSR is extensive and mature [49,50], giving polarization multiplexing an enormous potential in the field of OPA.

In 2019, Tyler NA proposed an OPA consisting of four sub arrays. By switching the working status of each sub array, the OPA achieved a longitudinal scanning range of 3° at the same input wavelength [51]. We proposed an OPA increasing the longitudinal scanning dimension through two subarrays with different grating periods as shown in Figure 4b [52]. The WGA is dual-layer fabricated on an Si-Si_3_N_4_ integration platform. Light is transmitted by Si waveguides, and the proportion of upward radiation is increased by the silicon nitride (Si_3_N_4_) grating above the waveguide. The OPA switches the working states of two subarrays by an optical switch, concatenating the longitudinal ranges of the two subarrays, and achieves a longitudinal scanning range of 32.6° through wavelength multiplexing within the light-source bandwidth range of 1500–1600 nm.

The structure of wavelength multiplexing OPA is simple, but it requires multiple sub arrays, resulting in a large chip size. Polarization multiplexing OPA can improve the utilization efficiency of the array. Combined with the mature design of PSR, polarization multiplexing OPA is a more efficient solution to expand the longitudinal scanning range of OPA.

**Figure 4 micromachines-15-00322-f004:**
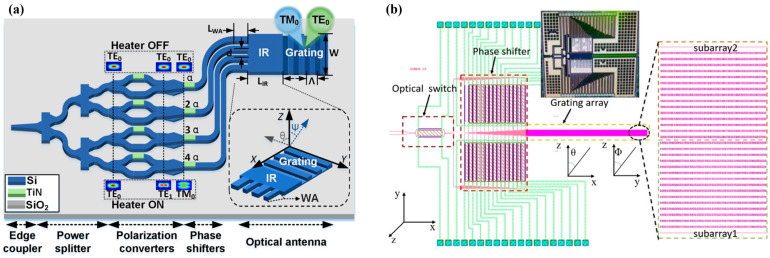
(**a**) Expanding the longitudinal scanning range of OPA by polarization multiplexing [47]. Opt. Express 30, 28,049 (2022). (**b**) Expanding the longitudinal scanning range of OPA by wavelength multiplexing [52]. Appl. Opt. 60, 5424 (2021).

#### 3.1.2. Reducing Divergence Angle of Longitudinal Dimension

According to Equation (2), it is necessary to extend the effective emission length of WGA to reduce the longitudinal divergence angle and improve the beam resolution [41,53,54,55]. For periodic WGA, the diffraction intensity exhibits an exponential decay, leading to a widening of the divergence angle. Designing apodized gratings to enhance the perturbation intensity along the WGA has been a mainstream scheme in recent years [41,54,56]. In 2021, Chen et al. designed two apodized gratings based on an Si-Si_3_N_4_ integration platform [41]. Offset etching was fabricated on the upper and lower surfaces of 340 nm thick Si_3_N_4_ to achieve the dual-level chain WGA and dual-level fishbone WGA. The etching width at both surfaces of the grating was expanded along the WGA to enhance the radiation intensity and ultimately achieved an effective emission length of 3 mm. In 2022, an apodized side-wall grating fabricated on 200 nm thick Si_3_N_4_ was demonstrated [54]. As shown in Figure 5a,b, from the beginning to the end of the grating, the width of inward etching gradually increases from 75 nm to 550 nm, achieving the aim of enhancing the radiation intensity. As can be seen from Figure 5c,d, the increase in perturbation intensity along WGA leads to uniformity of the radiation intensity. The effective emission length of the grating measured in the experiment was 3.16 mm, which was consistent with the simulation results. A longitudinal divergence angle of 0.04° was measured. 

Another scheme to extend the effective transmission length is to weaken the perturbation of the waveguide grating. In 2023, Qiu et al. proposed a method of shallow etching of 10 nm on the 220 nm Si surface and fabricated a 2 mm long WGA as shown in Figure 6 [55]. Gaussian apodization design was applied to the duty cycle of the grating along the propagation direction. The divergence angle measured in the experiment was 0.07°. In the same year, Luo et al. fabricated an OPA whose array was composed of periodic gratings [53]. The WGA was dual-layer with an Si waveguide and upper Si_3_N_4_ grating. The radiation intensity of the WGA theoretically decays exponentially with the propagation length because of the periodic structure, causing the actual effective length to be only 1.5 mm. Therefore, the experimental result of the longitudinal divergence angle was 0.05°, which is wider than the theoretical value of 0.016°. 

The length of the WGA needs to be designed reasonably to achieve the goal of reducing the longitudinal divergence angle. Blindly extending the length of the WGA may lead to other issues. Extending the effective emission length of the WGA requires reducing the perturbation intensity. When the perturbation intensity of the WGA decreases to a certain extent, the impact of manufacturing errors cannot be ignored, and may actually lead to an increase in the divergence angle. For apodized gratings, if the etching width and period cannot be precisely controlled along the WGA direction, it may also lead to a decrease in beam quality. Compared to Si, the refractive index contrast between Si_3_N_4_ and the cladding is smaller, having a larger process tolerance. The WGA with Si waveguide and Si_3_N_4_ grating has become a feasible solution for achieving longer emission length. Table 1 summarizes the OPA designed in recent years to improve the longitudinal beam performance.

### 3.2. Improving Beam Quality of the Transversal Dimension

#### 3.2.1. Distributing Array Periodically with Small Spacing to Expand the Transversal Scanning Range

The scanning range of the periodic OPA gradually expands as the channel spacing decreases [58,59,60]. In 2014, A. Yaacobi et al. reduced the channel spacing to 2 μm and a 16-channel periodic OPA was designed. The scanning angle of phase control was increased to 51° with a divergence angle of 3.3° [60]. With the channel spacing *d* < $\frac{\lambda }{2}$, the OPA can achieve non-aliasing beam scanning within a 180° range. At the same time, the crosstalk between adjacent channels increases with the reduction of the channel spacing. When the emission length of the grating is long, the crosstalk between adjacent channels will seriously affect the optical power of each channel. Therefore, it is an enormous challenge to suppress the crosstalk between the densely distributed optical phased array channels. In 2015, Song et al. proposed a superlattice structure consisting of five non-uniform waveguides [61]. The impact of the arrangement of waveguides with different widths in the superlattices on the suppression of crosstalk was analyzed in detail and the structure is shown in Figure 7a [62]. Specific sorting was performed on waveguides with unequal widths to enhance phase mismatch between adjacent channels and suppress directional coupling. Phare C T et al. proposed a 64-channel end face emitting OPA [63]. Waveguides were designed with different widths and cycled in groups of three. In this case, even if the channel spacing is 775 nm, low crosstalk between adjacent channels is achieved. Even when steered up to 60° off-axis, the single diffraction-limited beam can carry more than 72% of the power. Superlattice structures exhibit excellent performance in OPA emitted from the end facet [63,64], but face significant challenges when used for grating emission. In order to maintain consistent longitudinal radiation angles of each channel, the period of the grating needs to be adjusted according to the width of the waveguide [65]. It requires high precision in fabrication and is sensitive to manufacturing deviations. Sinusoidal silicon waveguides can be used to achieve ultra compact and low crosstalk OPA [66]. Liang et al. conducted research on densely distributed waveguides modulated by periodic bending, and an end facet emitting OPA with an FOV of 120° was achieved through sinusoidal waveguides in 2023 [67]. 

OPA with a plate interference coupling region was designed and an OPA with a shallow etched Si slab was demonstrated in 2020 [59,68]. The OPA had 32 channels with a periodic spacing of 1.65 mm and achieved a horizontal scanning range of 96°. First the light in each channel interferes with the other after entering the Si slab, and then radiates outward through the grating formed by a 70 nm shallow etching. The proposed antenna operates as a whole device to achieve beam steering without crosstalk and there is no need to optimize the width of the WGA to reduce crosstalk. In 2022, Liu et al. proposed a trapezoidal slab emission array with half-wavelength-pitch periodic channels, combining the advantages of superlattice waveguides and planar emission arrays [69]. The structure of superlattice waveguides was used in front of the trapezoidal slab to suppress crosstalk during waveguide convergence, achieving a dense channel spacing of 775 nm. Grating was formed by etching 10 nm downwards on the surface of a 220 nm thick Si flat plate. This OPA achieved a non-aliasing FOV of 180°, as shown in Figure 7b. 

**Figure 7 micromachines-15-00322-f007:**
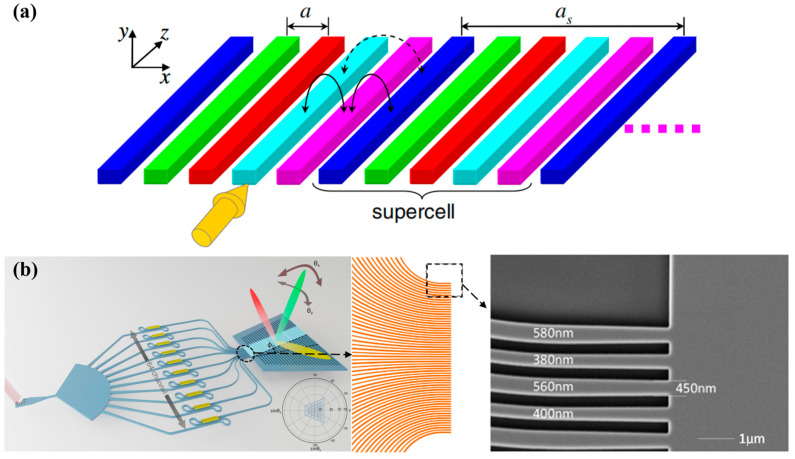
Densely distributed periodic OPAs. (**a**) The waveguide superlattice array [62]. Photonics Res. 4, 233 (2016). (**b**) OPA with a trapezoidal slab grating [69]. Optica 9, 903 (2022).

The combination of superlattice input waveguides and planar emission gratings can achieve periodic dense integration while avoiding the challenges posed by superlattice emission gratings to the process. Si_3_N_4_ gratings can be used, instead of the gratings formed by shallow etching of Si, to further extend the emission length. When the number of channels is small, the transversal aperture of the OPA with half-wavelength periodic distribution is not large, and the resolution of lateral scanning is limited as a result. For an OPA with small spacing and periodic distribution, in order to achieve a smaller lateral divergence angle, it is necessary to achieve a larger transversal aperture and increase the number of channels. In this case, how to achieve a densely integrated phase-shifting structure becomes a challenge.

#### 3.2.2. Distributing Array Unevenly with Large Spacing to Expand the Transversal Scanning Range

Another solution to expand the transversal scanning range is to distribute the arrays unevenly [43,44,70,71,72,73,74]. In 2009, A Hosseini et al. proposed a design method for a large angle beam steering OPA with non-equidistant emission units based on a silicon nanomembrane. The simulation results show that the array has a deviation angle of ±60 ° and ±45 ° in two directions, respectively [75]. In 2016, Intel Labs reported a 128-channel non-uniform OPA that achieved a transversal scanning angle of 80° and an average divergence angle of 0.14° [73]. The best resolution that could be measured was 0.11°, which was the highest resolution at that time. In 2021, a 128-channel large spacing aperiodic OPA based on genetic algorithm (GA) was demonstrated as shown in Figure 8a [44]. The objective function of the OPA far-field distribution optimization is set to a gate function at the maximum scan angle. The cosine similarity between the objective function and the actual intensity distribution is used as the optimization goal. The average spacing of channels is 29.7 μm. The transversal aperture of the array is 4 mm, achieving a divergence angle of 0.029°. The sidelobe level of this OPA is −10.2 dB at 0° and −6.9 dB at 70°. In 2023, this OPA was used to achieve wireless optical communication within a transversal range of 100° [9]. In 2022, Dong et al. proposed a scheme that adjusts the optical power amplitude of each channel by a variable optical attenuator (VOA), and then designed the array with non-uniform spacing, achieving sidelobe levels below −24.65 dB within an FOV of 120° [71]. The use of VOA is beneficial for modulating the amplitude of each channel, but also brings the disadvantage of reducing optical power, which poses a challenge for achieving long-distance detection. In 2023, Wang et al. fabricated a 256-channel aperiodic OPA achieving a transversal scanning range of 150°, as shown in Figure 8b [43]. The WGA was composed of two identical sidewall Si_3_N_4_ gratings with a displacement. GA was used to optimize the pitches for the non-uniform OPA and the transversal aperture was 1.8 mm [76]. The transversal divergence angle was 0.066° at *φ* = 0° and the side lobe suppression ratio (SLSR) remained above 2 dB at *φ* = ±75°. Table 2 summarizes the OPA designed in recent years to improve transversal beam performance.

Here we propose an optimization scheme for aperiodic OPA, which is based on the particle swarm optimization (PSO) algorithm. We take the influence of *f*(*φ*) on the intensity of the main beam during beam deflection into account in the optimization step. This scheme can predict the actual test results of beam steering to a certain extent. The scheme is combined with the directional factor *f*(*φ*) of the WGA, the number of channels N, and the scanning range to be optimized (*φ_min_*, *φ_max_*). The objective function is determined by the maximum and minimum of the SLSR within the scanning range as a measure of overall performance within the scanning range. Assuming the direction function of a WGA is *f*(*φ*) = cos2(φ), *φ*
$\in [-\frac{\pi }{2},\frac{\pi }{2}]$, the profile factor of each emitter in OPA has a full width at half-maximum (FWHM) of 90°. By substituting the formula of *f*(*φ*) into Equation (4) and combining it with the positions of each channel, the light intensity distribution of OPA at various angles in the far field can be calculated. SLSR is used to measure the ability to distinguish between the main beam and the sidelobes. The objective function of the PSO algorithm is the product of the maximum and the minimum of SLSR within the predetermined scanning range. The larger the value of the product, the stronger the ability to distinguish the main beam within the global range of the predetermined scanning range. 

The algorithm proposed above was used to optimize a 128-channel OPA within ±70° with the channel spacing of 10–25 μm. The optimized OPA has a transversal aperture of 2.17 mm with an average spacing of 17.11 μm. Figure 9a shows the intensity distribution of the OPA when the main beam deviates. The maximum SLSR is 13.5 dB at 0° and the divergence angle of the main beam is 0.036°, as shown in Figure 9b. The SLSR is better than 4.5 dB in the whole FOV of 140°. Figure 9c shows the channel spacing for 128 channels.

### 3.3. Unidirectionality of WGA and OPA Chips

To satisfy the requirement of long-distance detection, OPA needs to have a high output optical power [79]. One of the key factors affecting the output optical power is the unidirectionality *D* of the emission array. The higher the proportion of upward radiation optical power in the WGA, the greater is the output optical power of the OPA. To enhance the unidirectionality of the WGA, the refractive index of the material in the vertical direction can be changed to increase the proportion of upward radiation [79,80,81]. On the other hand, utilizing the phase interference of radiated light in the upper and lower directions can achieve a higher proportion of upward radiation [39,41,43,56,82]. In 2017, Raval et al. proposed a dual-layer WGA with displacement. By utilizing constructive interference and destructive interference of light in the WGA as shown in Figure 10a,b, unidirectionality over 90% could be achieved [56]. A unidirectionality over 92% was obtained by testing the output power of the OPA [43]. In 2021, a dual-level chain WGA and a dual-level fishbone WGA by etching on both the upper and lower surfaces with an offset on a single Si_3_N_4_ layer [41], as shown in Figure 10c,d, achieved the unidirectional of 95%. 

Compared to changing the index in the vertical direction of the material system to enhance the proportion of upward radiation, obtaining higher unidirectionality by utilizing the phase interference of radiated light in the upper and lower directions can maintain the perturbation intensity of WGA and will not decrease the effective emission length. In addition, Si_3_N_4_ can withstand higher input optical power compared to Si [83]. Compared to Si_3_N_4_, Si has a higher refractive index contrast and a great potential in preparing longer gratings. Fabricating OPA on Si-Si_3_N_4_ integrated platforms is a major trend. Table 3 lists the research results on the unidirectionality of WGA in recent years.

## 4. Conclusions

PIC OPA has great potential in achieving solid-state LiDAR due to its compatibility with mature CMOS processes. This article summarizes the schemes to improve the beam quality and device performance of the OPA conducted in recent years. To improve the resolution of the beam, it is necessary to design a longer WGA and larger lateral apertures. The advantage of high input power affordability, low transmission loss of Si_3_N_4_, and small-spacing dense integration of Si can be combined on an Si-Si_3_N_4_ integration platform. The Si-Si_3_N_4_ dual-layer WGA, which transmits light by Si waveguides and radiates light outward by Si_3_N_4_ gratings, can effectively extend the emission length. At the same time, due to the higher refractive index and smaller cross-sectional dimensions of Si waveguides, smaller spacing and larger scale OPA arrays can be achieved. Therefore, developing an OPA on the Si-Si_3_N_4_ platform is an effective way to achieve large-aperture OPA. 

When the period spacing of a periodic OPA is larger than half-wavelength, the scanning range will be limited. Designing an OPA with a half wavelength periodic distribution can achieve 180° aliasing free scanning. In order to reduce the divergence angle and improve the resolution, it is necessary to expand the lateral aperture and increase the number of channels. When integrating on a large scale, reasonable layout phase shifters and other devices as well as reducing chip size become the major challenges for fabrication in the future. Non-periodic OPA is an effective solution for achieving a large aperture and large scanning range. An optimization scheme for channel spacing of aperiodic OPA has been proposed. Non-periodic OPA with large spacing cannot eliminate the presence of grating lobes, resulting in the dispersion of optical power into various side lobes. How to further reduce the SLSR and increase the energy proportion of the main beam warrant further research.

Utilizing the interference effect of the phase of light in the channels to achieve high unidirectionality and output optical power is an effective solution. Coherent light ranging and frequency modulated continuous wave ranging by OPA remains a current research hotspot. With the small size, light weight, and high-quality beam scanning, OPA will have great prospects in the fields of all-solid-state LiDAR, wireless optical communication, and imaging.

## Figures and Tables

**Figure 1 micromachines-15-00322-f001:**
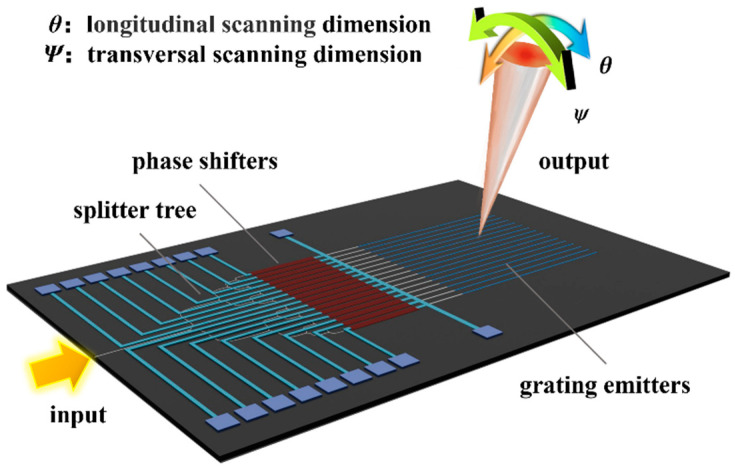
Schematic of OPA and 2D beam-steering.

**Figure 2 micromachines-15-00322-f002:**
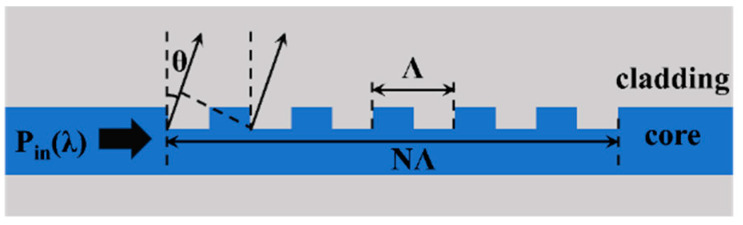
Schematic of the waveguide grating structure.

**Figure 3 micromachines-15-00322-f003:**
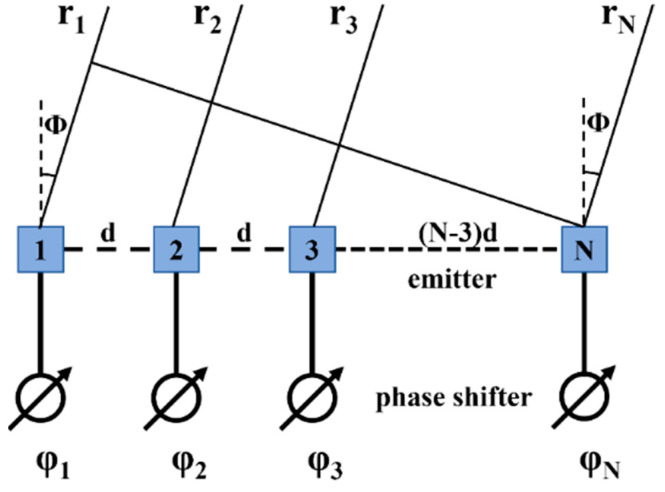
A 1-D OPA with N units.

**Figure 5 micromachines-15-00322-f005:**
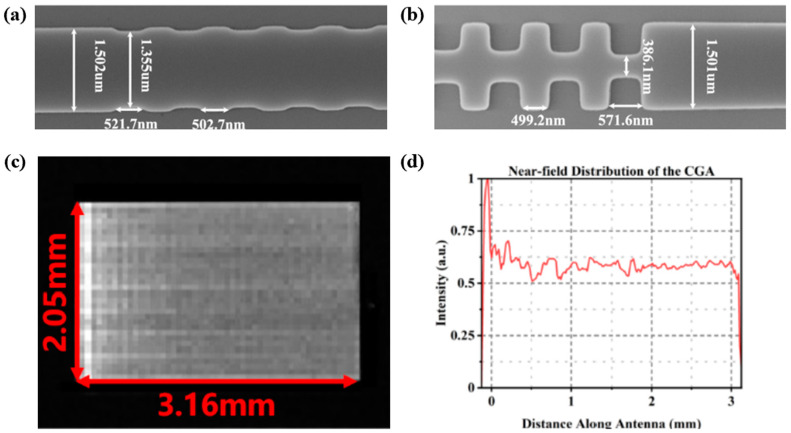
Apodized side-wall WGA fabricated on a 200 nm thick Si_3_N_4_ film [54]. Opt. Express 30, 28112 (2022). (**a**,**b**) The beginning and the end of WGA. (**c**,**d**) IR camera image of near-field radiation of OPA and normalization curve of the radiation along the WGA.

**Figure 6 micromachines-15-00322-f006:**
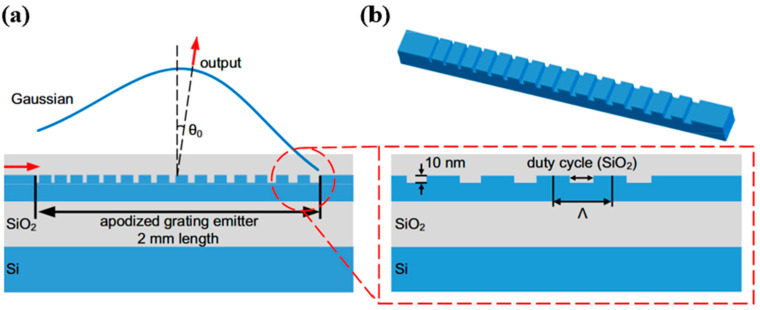
(**a**)The Gaussian distribution design of the output optical power of the grating in the longitudinal dimension. (**b**) The 10 nm shallow etched Si grating [55]. Photonics Res. 11, 659 (2023).

**Figure 8 micromachines-15-00322-f008:**
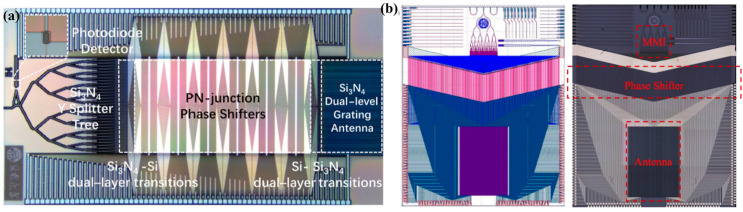
OPAs with non-equidistant channels. (**a**) A 128-channel aperiodic OPA designed by [44]. Photonics Res. 9, 2511 (2021). (**b**) A 256-channel aperiodic OPA designed by [43]. Opt. Express 31, 21192 (2023).

**Figure 9 micromachines-15-00322-f009:**
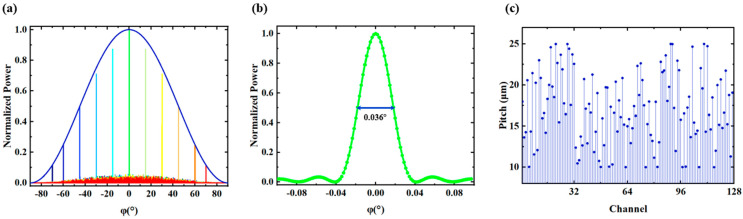
(**a**) Far-field simulation result with the main beam at different angles (0°, ±15°, ±30°, ±45°, ±60°, ±70°) in *φ*-dimension. (**b**) With the main beam at *φ* = 0°, the light intensity distribution is within ±0.1°. The divergence angle of the main beam is 0.036°. (**c**) Pitch distribution of 127 channel spacings.

**Figure 10 micromachines-15-00322-f010:**
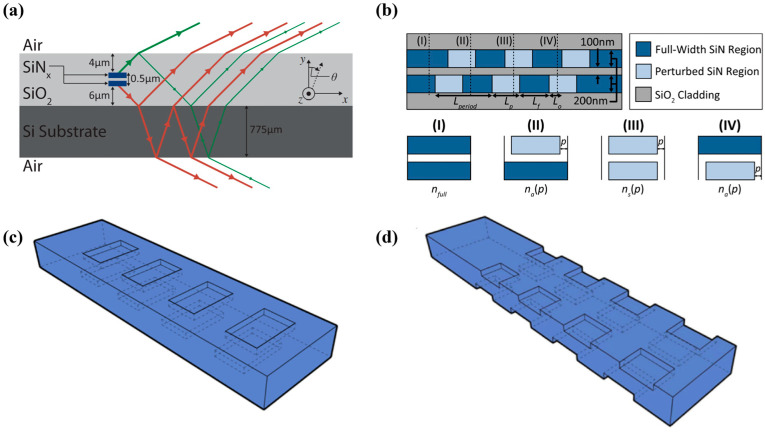
(**a**,**b**) Dual-layer WGA [56]. Opt. Lett. 42, 2563 (2017). (**c**,**d**) Dual-level chain WGA and dual-level fishbone WGA [41]. Opt. Express 29, 20995 (2021).

**Table 1 micromachines-15-00322-t001:** OPA for improving the longitudinal beam performance.

	Ref	Platform	Scheme	Divergence	Length of WGA	Scanning Rang
Expanding scanning range	[45]	Si	Polarization Multiplexed	/	/	28.2°
	[46]	Si	Polarization Multiplexed	/	/	54.5°
	[54]	Si	Polarization multiplexed	/	/	15.62° + 16.08°
	[48]	Si	Polarization multiplexed	2.0°	/	28°
	[52]	Si-Si_3_N_4_	Wavelength multiplexing	0.067°	/	32.6°
Reducing longitudinal divergence angle	[56]	Si_3_N_4_	Apodized WGA	/	3 mm	/
	[57]	Si	Subwavelength WGA	0.081°	1 mm	0.17°/nm
	[41]	Si_3_N_4_	Apodized WGA	0.029° [44]	3 mm	/
	[54]	Si_3_N_4_	Apodized WGA	0.04°	3.16 mm	0.064°/nm
	[55]	Si	Gaussian apodized WGA	0.07°	2 mm	13.2°
	[53]	Si- Si_3_N_4_	Uniform periodic WGA	0.05°	5 mm	15.1°

**Table 2 micromachines-15-00322-t002:** OPA for improving transversal beam performance.

Ref	Platform	Scheme	Divergence Angle	Number of Channels	Scanning Range
[60]	Si	Periodic	3.3°	16	51°
[63]	Si	Periodic	1.2°	64	120°
[77]	Si-Si_3_N_4_	Periodic	0.69°	64	35.5°
[78]	Si-Si_3_N_4_	Periodic	1.9°	64	96°
[53]	Si-Si_3_N_4_	Periodic	0.04°	1024	40°
[68]	Si-Si_3_N_4_	Si slab array	2.3°	32	96°
[69]	Si	Si slab array	2.1°	64	180°
[73]	Si	Aperiodic	0.14°	128	80°
[44]	Si-Si_3_N_4_	Aperiodic	0.021°	128	140°
[43]	Si_3_N_4_	Aperiodic	0.066°	256	150°
[72]	Si-Si_3_N_4_	Aperiodic	0.051°	256	140°

**Table 3 micromachines-15-00322-t003:** Research on unidirectionality of OPA.

	Ref	Platform	Scheme	Unidirectionality
Simulation	[79]	Si	Etching upper SiO_2_ cladding	>70%
	[39]	Si-Si_3_N_4_	Shallow etching Si_3_N_4_ grating	>89%
	[59]	Si-poly-Si	High contrast grating	93.94%
	[82]	Si-Si_3_N_4_	Interleaved etching of Si_3_N_4_ grating	97%
Fabrication and testing	[56]	Si_3_N_4_	Dual-layer Si_3_N_4_ grating with offset	>90%
	[41]	Si_3_N_4_	Etching dual-level Si_3_N_4_ grating with offset	~80–90%
	[43]	Si_3_N_4_	Dual-layer Si_3_N_4_ grating with offset	>92%

## Data Availability

Data are contained within the article.

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
