# Peer review of "Beam Steering Technology of Optical Phased Array Based on Silicon Photonic Integrated Chip"

_micromachines, 2024, doi:10.3390/mi15030322_

Round 1
Reviewer 1 Report
Comments and Suggestions for Authors
The authors report on progress of the beam steering technology of optical phased array and propose an optimization scheme for aperiodic OPA in this paper. This article is well written with clear logic and sufficient research. However, it requires minor revisions before ready for publication.
Minor comments:
1. There are some errors or omissions in the sorting of figures in this paper, including Fig.6-Fig.10.
2. The reference is missing in Fig.6 and the Fig. 7 (b) needs to be enlarged.
3. The subscript format of Si3N4 is incorrect.
4. There needs to be a space between numbers and units.
5. The format of (a) and (b) in Fig. 8 and Fig. 9 need to be consistent with others.
6. Some of the references is incorrect in format, such as[6],[12],etc. The author needs to check it carefully.
Comments on the Quality of English LanguageThe English writing is qualified with a few minor revisions.
Author Response
|
Response to Reviewer 1 Comments
|
||
|
1. Summary |
|
|
|
Thank you very much for taking the time to review this manuscript. Please find the detailed responses below and the corresponding revisions highlighted changes in the re-submitted files.
|
||
|
2. Questions for General Evaluation |
Reviewer’s Evaluation |
Response and Revisions |
|
Is the work a significant contribution to the field? |
|
[Please give your response if necessary. Or you can also give your corresponding response in the point-by-point response letter. The same as below] |
|
Is the work well organized and comprehensively described? |
|
|
|
Is the work scientifically sound and not misleading? |
|
|
|
Are there appropriate and adequate references to related and previous work? |
|
|
|
Is the English used correct and readable? |
|
|
3. Point-by-point response to Comments and Suggestions for Authors |
|||
Comments 1: There are some errors or omissions in the sorting of figures in this paper, including Fig.6-Fig.10. |
|||
Response 1: Thank you for pointing this out. We agree with this comment. Therefore, I made a modification on line 169, line 197, line 221, line 231,etc of the attachment and highlighted it in red font.
|
|||
Comments 2: The reference is missing in Fig.6 and the Fig. 7 (b) needs to be enlarged. |
|||
Response 2: Agree. We have added the cited literature in Fig.6 on page 6, line 181, and enlarged the details of image 7 (b) on page 8, line 224. The above modifications are highlighted in red font.
|
|||
Comments 3: The subscript format of Si3N4 is incorrect. |
|||
Response 3: Agree. We have corrected the incorrect subscripts and used red font. I believe you can easily see these corrections. |
|||
Comments 4: There needs to be a space between numbers and units. |
|||
Response 4: Agree. We added spaces between numbers and units and used red font. I believe you can easily see these corrections. |
|||
Comments 5: The format of (a) and (b) in Fig. 8 and Fig. 9 need to be consistent with others. |
|||
Response 5: Agree. We have made modifications to the format of Fig.8 and Fig.9. However, it is not convenient to use red font to highlight the image. Please pay attention to the changes in the pictures. |
|||
Comments 6: Some of the references is incorrect in format, such as[6],[12],etc. The author needs to check it carefully. |
|||
Response 6: Agree. The citation format is incomplete. Page numbers are missed. We have corrected the correct format and highlighted it in red font in the attachment. |
|||
4. Response to Comments on the Quality of English Language |
|||
Point 1: The English writing is qualified with a few minor revisions. |
|||
Response 1: Thank you for your suggestion. We have checked and revised the resubmitted manuscript. |
|||
5. Additional clarifications |
|||
The parts highlighted in red in the resubmitted manuscript are all modified parts. Some of the modifications come from the comments of other reviewers. |
|||
Reviewer 2 Report
Comments and Suggestions for Authors
This is a review of the beam steering technology by optical phased array based on silion photonic chip. The paper is very meaningful for learning about the OPA technology. I suggest this paper to be accepted for publication in micromachines, giving below enhancements.
1.The keywords are not suitable and need to be modified.
2.There should be spaces before numbers and units, like mm and dB.
3.Please use the same format for the same symbol. The symbols need to be in italics, like L and d.
4.The subscript is not standardized. Like TE0, TM0, Si3N4.
5.The authors should list some references for the equations used in section 2.
6. There is an typo in formula 4.
7.The description of the optimization scheme for aperiodic OPA shall be in more detail.
This is a review of the beam steering technology by optical phased array based on silion photonic chip. The paper is very meaningful for learning about the OPA technology. I suggest this paper to be accepted for publication in micromachines, giving below enhancements.
1.The keywords are not suitable and need to be modified.
2.There should be spaces before numbers and units, like mm and dB.
3.Please use the same format for the same symbol. The symbols need to be in italics, like L and d.
4.The subscript is not standardized. Like TE0, TM0, Si3N4.
5.The authors should list some references for the equations used in section 2.
6. There is an typo in formula 4.
7.The description of the optimization scheme for aperiodic OPA shall be in more detail.
Author Response
For review article
|
Response to Reviewer 2 Comments
|
||
|
1. Summary |
|
|
|
Thank you very much for taking the time to review this manuscript. Please find the detailed responses below and the corresponding revisions highlighted changes in the re-submitted files.
|
||
3. Point-by-point response to Comments and Suggestions for Authors |
|||
Comments 1: The keywords are not suitable and need to be modified. |
|||
Response 1: Thank you for pointing this out. We checked the keywords and found that one of them was not suitable, so we deleted ‘diffraction’ and kept the rest. |
|||
Comments 2: There should be spaces before numbers and units, like mm and dB. |
|||
Response 2: Agree. We added spaces between numbers and units and used red font. We use red font as the annotation at line 117,line 118,line 121 and so on. I believe you can easily see these corrections.
|
|||
Comments 3: Please use the same format for the same symbol. The symbols need to be in italics, like L and d. |
|||
Response 3: Agree. We have adjusted the font of the symbols to italics and used red font. We use red font as the annotation at line 70,line 72,line 73 and so on. I believe you can easily see these corrections.
|
|||
Comments 4: The subscript is not standardized. Like TE0, TM0, Si3N4. |
|||
Response 4: Agree. We have corrected the incorrect subscripts and used red font. We use red font as the annotation at line 116,line 119,line 127 and so on. I believe you can easily see these corrections. |
|||
Comments 5: The authors should list some references for the equations used in section 2. |
|||
Response 5: Agree. We have added references to the formulas mentioned in the manuscript. Among them, Eq (4) is the derivation result of a thesis. Eq. (4) has different expressions and there is no completely consistent literature source. |
|||
Comments 6: There is an typo in formula 4. |
|||
Response 6: I didn't realize where the mistake was. I guess you think the error lies in r0 and ri, so I added an explanation in red font on lines 93-96. If my understanding is incorrect, please do not hesitate to give me advice. |
|||
Comments 7 : The description of the optimization scheme for aperiodic OPA shall be in more detail. |
|||
Response 7: We have added a paragraph in red font at lines 270-278 to explain our proposed optimization algorithm. |
|||
4. Response to Comments on the Quality of English Language |
|||
Point : This is a review of the beam steering technology by optical phased array based on silion photonic chip. The paper is very meaningful for learning about the OPA technology. I suggest this paper to be accepted for publication in micromachines, giving below enhancements. |
|||
Response : The comments in this section are the same as the previous section. Please refer to the previous section for detailed answers. |
|||
5. Additional clarifications |
|||
The parts highlighted in red in the resubmitted manuscript are all modified parts. Some of the modifications come from the comments of other reviewers. |
|||
Reviewer 3 Report
Comments and Suggestions for Authors
The authors provide a review of recent progress in the technology of Optical Phased Arrays to be used in LIDARS. I found the work useful and timely. I would be glad to recommend it for publication, except for one reservation. The manuscript part around line 250, as well as Summary describes a newly proposed by the authors, that presumably was not published previously. For a review paper, I would suggest to limit the original idea to a short comment and descibe the new result in a separate publication.
Comments on the Quality of English LanguageEnglish requires some polishing
Author Response
Dear Reviewer,
Thank you for your suggestion. We think that this work is not yet comprehensive enough, and publishing it as a review is somewhat lacking. We have applied this work to the preparation of OPA, and we will verify this work in subsequent experiments. Afterwards, we will consider publishing the results of this work.
The parts highlighted in red in the resubmitted manuscript are all modified parts. Some of the modifications come from the comments of other reviewers.
Best regards,
Jinyu Wang